# COMFYGEN: PROMPT-ADAPTIVE WORKFLOWS FOR TEXT-TO-IMAGE GENERATION

**Rinon Gal**
NVIDIA, Tel Aviv University

**Adi Haviv**
Tel Aviv University

**Yuval Alaluf**
Tel Aviv University

**Amit H. Bermano**
Tel Aviv University

**Daniel Cohen-Or**
Tel Aviv University

**Gal Chechik**
NVIDIA

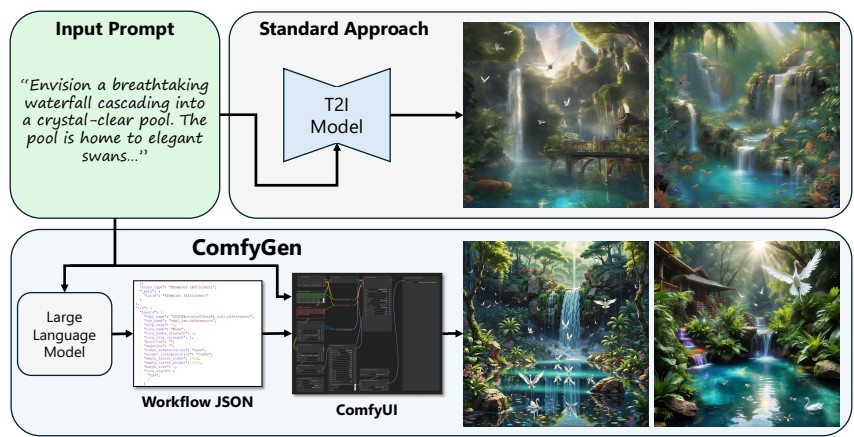

Figure 1: The standard text-to-image generation flow (top) uses a single monolithic model to transform a prompt into an image. However, the user community often relies on complex workflows with specialized components, hand-crafted by expert users for different scenarios. We leverage an LLM to automatically synthesize such workflows, conditioned on the user's prompt (bottom).

## ABSTRACT

The practical use of text-to-image generation has evolved from simple, monolithic models to complex workflows combining multiple specialized components. These components are independently trained by different practitioners to excel at specific tasks – from improving photorealism or anime-style generation to fixing common artifacts like malformed hands. Using these components to craft effective workflows requires significant expertise due to the large number of available models and their complex interdependencies. We introduce prompt-adaptive workflow generation, where the goal is to automatically tailor a workflow to each user prompt by intelligently selecting and combining these specialized components. We propose two LLM-based approaches: a tuning-based method, and an in-context approach. Both approaches lead to improved image quality compared to monolithic models or generic workflows, demonstrating that prompt-dependent flow prediction offers a new pathway to improving text-to-image generation.

## 1 INTRODUCTION

Recent advances in text-to-image generation led to a shift from simple, monolithic workflows to more complex ones that combine multiple specialized components. The community has produced a rich ecosystem of independently trained components, each designed to address specific aspects of image generation: fine-tuned models optimized for photorealism, for anime-style generation, or for specific subject matter; LoRAs trained to correct anatomical issues like malformed hands or facial features; improved latent decoders for enhanced detail; and super-resolution blocks for various artistic styles. These components are often developed and trained by different practitioners, each focusing on solving particular challenges in image generation. When combined effectively, this diverse collection of specialized components offers significant potential for improving generation quality. Importantly, effective workflows are prompt-dependent, with the optimal choice of components often depending on the content being generated. For example, workflows for nature photographs may

benefit from photorealism-focused models and texture-enhancing upscalers, while those for human images often require specific anatomical corrections. However, due to the complexity of available components and their interactions, building well-designed workflows typically requires considerable expertise in understanding how different specialized components can complement each other.

## 2 METHOD

We propose to leverage LLMs to construct text-to-image generation workflows conditioned on user prompts. We present two approaches: ComfyGen-IC: Uses Claude Sonnet 3.5 with in-context learning to select workflows based on a table of flow performances across different categories. The LLM analyzes new prompts and matches them to flows that performed well on similar content. ComfyGen-FT: Fine-tunes Llama 3.1 (Dubey et al., 2024) to predict effective flows given a prompt and target user-preference score. To train these models, we collect 500 diverse prompts and generate images using 310 different flows from a popular model sharing website[1]. The images are scored using an ensemble of aesthetic predictors and human preference estimators (Kirstain et al., 2023; Xu et al., 2024; Wu et al., 2023). This dataset of (prompt, flow, score) triplets captures how different component combinations perform across various generation scenarios, and is used for fine-tuning.

## 3 RESULTS

We compare ComfyGen to three types of approaches: (1) Single model approaches (base SDXL (Podell et al., 2024), popular fine-tunes, DPO-optimized versions (Wallace et al., 2024)) (2) Fixed, popular workflows, and (3) Other uses of LLMs to improve generation through layout prediction or repeated-editing (Zhenyu et al., 2024; Yang et al., 2024).

On both automatic metrics (GenEval (Ghosh et al., 2024) on their standard benchmark, HPS V2.0 (Wu et al., 2023) on 500 test prompts) and user studies (two alternative forced-choice on

| Model | Single object | Two object | Counting | Colors | Position | Attribute binding | Overall |
|---|---|---|---|---|---|---|---|
| SD2.1 | 0.98 | 0.51 | 0.44 | 0.85 | 0.07 | 0.17 | 0.50 |
| SDXL | 0.98 | 0.74 | 0.39 | 0.85 | 0.15 | 0.23 | 0.55 |
| JuggernautXL | **1.00** | 0.73 | 0.48 | 0.89 | 0.11 | 0.19 | 0.57 |
| DreamShaperXL | 0.99 | 0.78 | 0.45 | 0.81 | 0.17 | 0.24 | 0.57 |
| DPO-SDXL | 1.00 | 0.81 | 0.44 | **0.90** | 0.15 | 0.23 | 0.59 |
| GenArtist | 0.94 | 0.41 | 0.40 | 0.72 | **0.24** | 0.07 | 0.47 |
| RPG-DiffusionMaster | **1.00** | 0.64 | 0.21 | 0.89 | 0.20 | **0.35** | 0.55 |
| Most Popular Flow | 0.95 | 0.38 | 0.26 | 0.77 | 0.06 | 0.12 | 0.42 |
| 2nd Most Popular Flow | **1.00** | 0.65 | **0.56** | 0.86 | 0.13 | 0.34 | 0.59 |
| ComfyGen-IC (ours) | 0.99 | 0.78 | 0.38 | 0.84 | 0.13 | 0.25 | 0.56 |
| ComfyGen-FT (ours) | 0.99 | **0.82** | 0.50 | **0.90** | 0.13 | 0.29 | **0.61** |

Figure 2: GenEval (Ghosh et al., 2024) comparisons. ComfyGen-FT outperforms all baseline approaches. SD2.1 results provided for calibration.

pairs sampled from the 500 test prompts, with 892 responses from 38 users), ComfyGen demonstrates superior performance by selecting components that better match the generation task. Qualitative results are in the appendix.

## 4 CONCLUSION

We demonstrate that automatically constructing prompt-dependent workflows from existing components offers a new path to improving text-to-image generation quality. This approach leverages the rich ecosystem of independently developed and tuned components, combining them in ways that best serve each generation request. Future work could explore expanding this to image-to-image tasks and enabling interactive workflow refinement through user feedback.

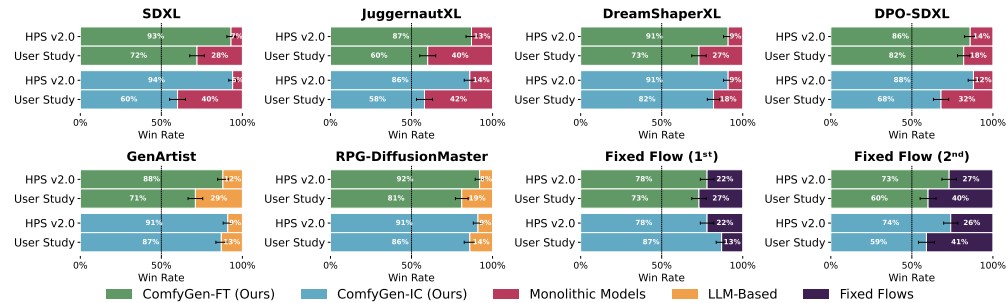

Figure 3: HPS V2.0 and User Study win rates. We compare each baseline against both ComfyGen-FT (green) and ComfyGen-IC (teal). ComfyGen variants are favored over all baselines.

[1] civitai.com

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

# COMFYGEN: PROMPT-ADAPTIVE WORKFLOWS FOR TEXT-TO-IMAGE GENERATION - APPENDIX

## A    QUALITATIVE RESULTS

In figs. 4 and 5, we provide a few qualitative examples of images generated with our approach, using SDXL-level models. In fig. 6 we show a comparison against selected baselines on GenEval prompts.

## B    WORKFLOW REPRESENTATION

To represent and run our flows, we leverage ComfyUI, an open-source software for designing and executing generative pipelines. In ComfyUI, users create pipelines by connecting a graph of blocks that represent specific models and their parameter choices. These include blocks for loading models, specifying prompts and latent dimensions, but also VAE decoders, LoRAs (Ryu, 2023), learned embeddings (Gal et al., 2022), ControlNets (Zhang et al., 2023), IP-Adapters (Ye et al., 2023), blocks that re-write and enhance the input prompt, super resolution models and more. Importantly, ComfyUI pipelines can be exported to a JSON file which outlines both the graph nodes and their connectivity. Our approach predicts this JSON format.

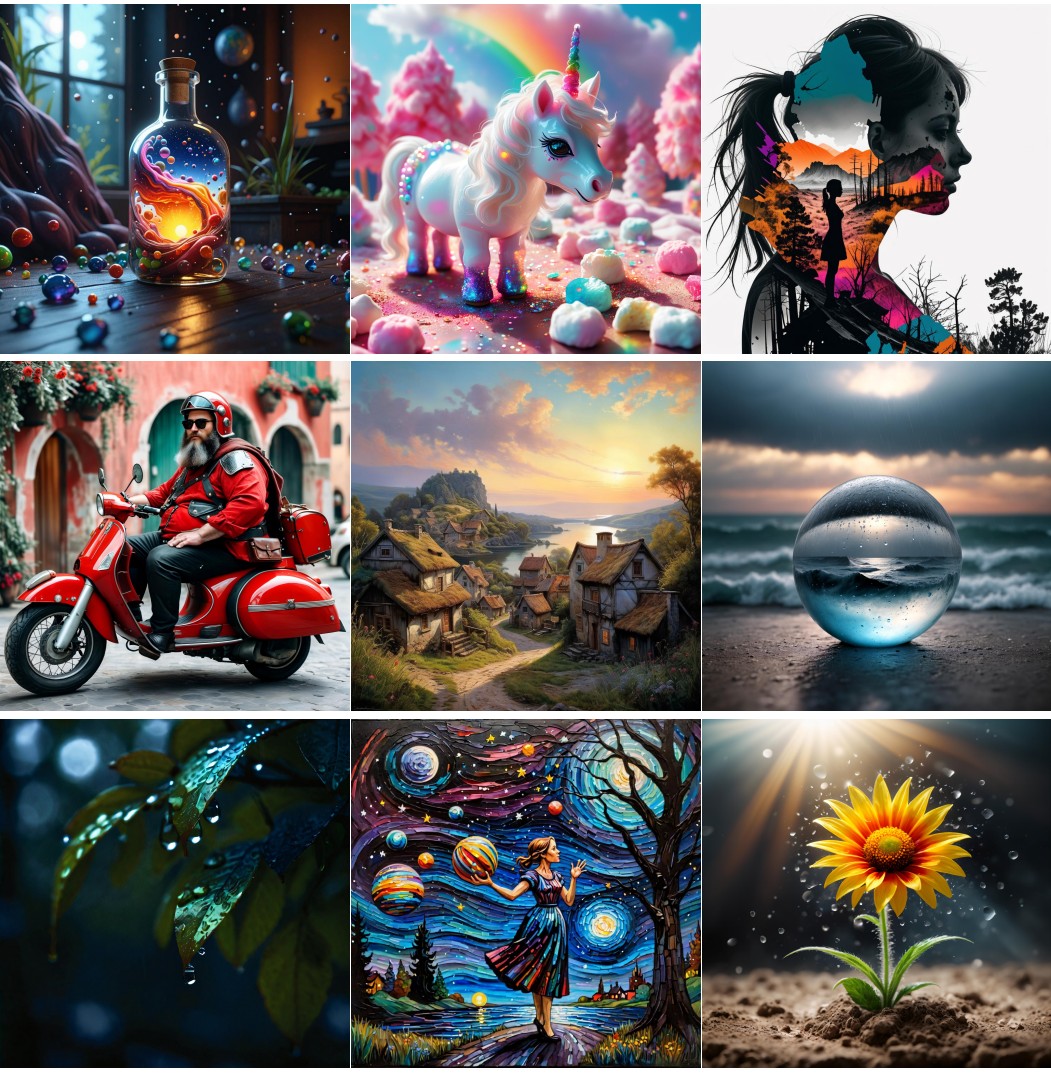

Figure 4: Our method can generate higher quality images across diverse domains and styles.

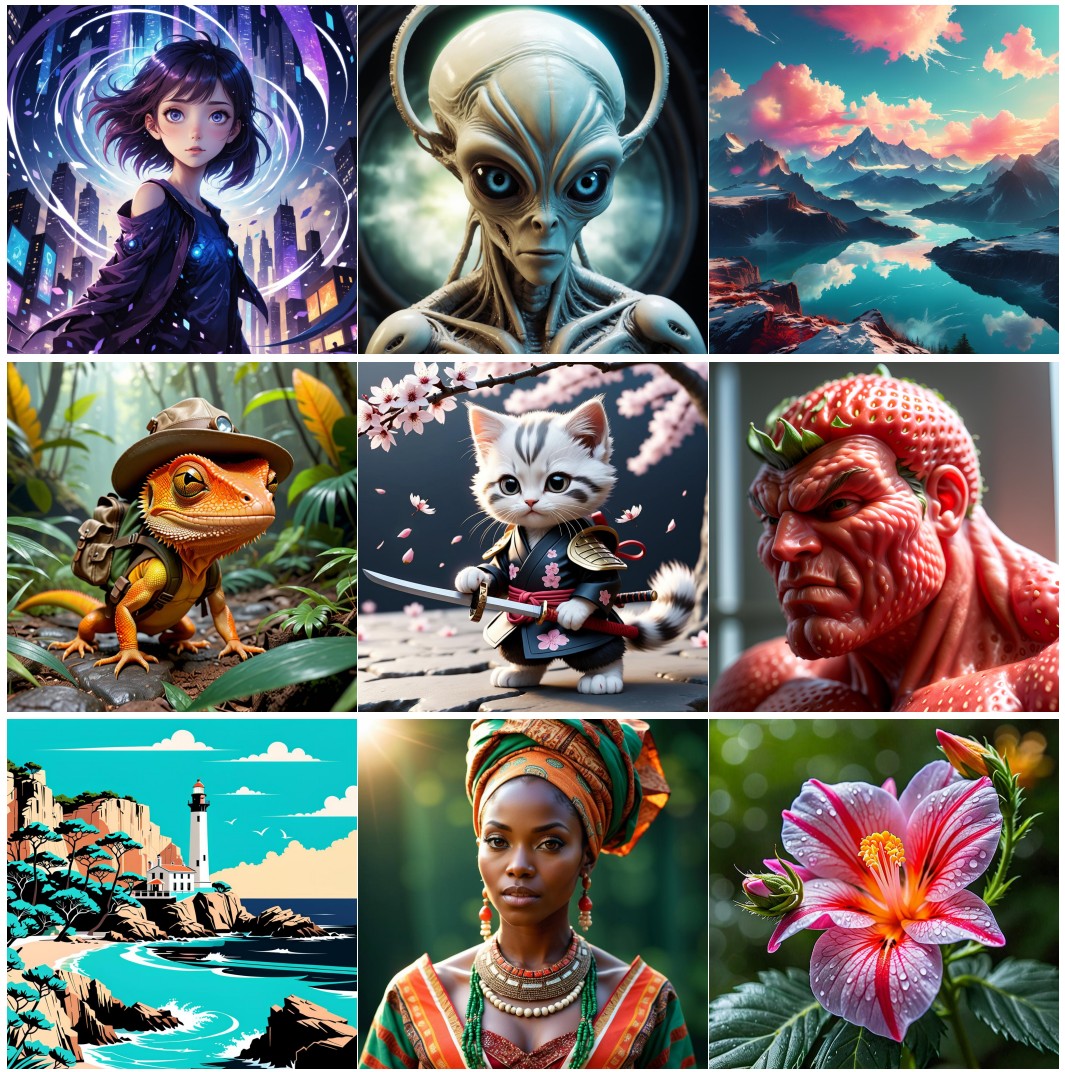

Figure 5: Additional qualitative results.

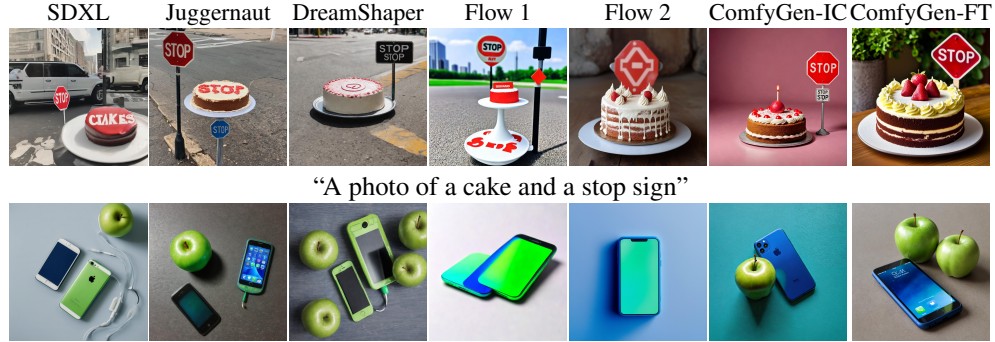

"A photo of a cake and a stop sign"

"A photo of a blue cell phone and a green apple"

Figure 6: Qualitative comparisons against selected methods on GenEval prompts.

