# OpenReview forum: "ComfyGen: Prompt-Adaptive Workflows for Text-to-Image Generation"
_ICLR.cc/2025/Workshop/MCDC — MCDC @ ICLR 2025_

### Official Review · Reviewer_hxGf · 2025-02-25

**Rating:** 4
**Confidence:** 4
**Fit:** 1

**Summary:**

This work introduces ComfyGen, an LLM-based component introduced at the beginning of the ComfyUI system (* additional detail), that is able to generate the optimal ComfyUI workflow configuration for a given input prompt. This addition greatly simplifies the procedure of selecting the optimal combination of modular components into a workflow for a given prompt, which typically requires significant expertise and experimentation.  The authors evaluate their contribution in an end-to-end fashion, comparing their system on the GenEval benchmark against other single model systems and fixed workflow configurations. The addition of their per prompt optimised workflow generation is able to produce images that seem to convincingly beat both single models, and fixed generation workflows in the ComfyUI pipeline results. Additionally, the authors show through a user study that the images produced with the introduction of ComfyGen are able to outperform their baseline results.

*an open-source system  addressing the problem of modularity in T21 problems by allowing users to create (T21) generation pipelines using a graph-based interface; in effect, this allows users to wire together monolithic components into a larger workflow to create more impressive results than each of the systems in isolation.

**Reason For Giving A Higher Score:**

The idea of the paper, in isolation, is very interesting and a fantastic addition to the ComfyUI workflow. Provided it works well, it would provide extreme benefit for the end users of this system.

**Reason For Giving A Lower Score:**

The paper’s relevance to the overall workshop theme is marginal.
Missing information in the paper puts into question the veracity of the claims.

**Strengths And Weaknesses:**

In short: this paper presents a very interesting mechanism to tie together monolithic components into a text-to-image generation workflow on top of ComfyUI. However, the specific contribution of the paper is not related to the MCDC workshop CFP in its entirety; the contribution is essentially the development of an LLM that is able to predict structured JSON output for the ComfyUI system based on a text prompt. This is not modular, collaborative nor decentralised; in effect it is an add-on to an existing system that enables modularity.  Furthermore, there are many details that have been omitted from the paper which leads one to doubt the claims made in the paper. Specific details are below:

**Strengths:**

- The idea of ComfyGen seems absolutely necessary for the proper use of the ComfyUI. In isolation, the idea is very compelling if one assumes the significance of the results presented without the additional detail that has been requested below. It seems that it will provide great benefit for those who wish to use the system for T2I generation.
- Overall, great presentation of the idea. The introductory figure explains the idea of the paper well, and the work is largely well presented.
- Great balance between automatic metrics and user study preference: this shows that this work brings improvement to the end user beyond what is quantified in GenEval. Particularly, Figure 3 is well presented.

**Weaknesses:**
- Not related to the scope of the MCDC CFP. The authors' contribution is the development of an LLM that predicts structured JSON output for ComfyUI based on textual input. Yes, this then ties together individual components in a modular way with ComfyUI. However, the contribution itself is more to LLM modeling than the MCDC CFP.
- More details on ComfyUI would be useful to contextualise the work.
 - Very little details on their experiment setup related to the creation of ComfyGen-IC and ComfyGen-FT which would prevent its reproduction. Specifically:
    - How was the data curated? What types of models did you consider? How were the workflows created? Additional details besides the summary statistics would be helpful to contextualise the work.
    - IC Learning: How did you structure the prompt?
    - FT: What was the training specification? For how many epochs did training occur ? Which optimiser? What was the learning rate?
- Results in the third section:
    - The results were gathered over one seed. This is not statistically significant.
    - How were the other baselines reproduced?
    - Ambiguity on the workflow baselines: what is the most popular workflow? Additional detail would be helpful.
- Limited details on the user study: How many participants? How were the participants recruited? How did you run the study?

NB: It is noted that the 2 page limit of the submission does inherently contribute to the brevity in details of any submitted work. However, the lack of detail in the correct paper version is viewed to be independent from this content restriction. In its current form, the work is not recommended for acceptance.

**Suggestions:**

Section 1 or 2:
Introducing ComfyUI more specifically here and not in the appendix would be useful. The contribution builds directly on top of ComfyUI.

Section 3:
Elaborate on the workflow baselines: what are the most popular workflows? As per the weaknesses, please provide more details about the experiment setup so that it would be reproducible. Why/How did you choose the specific baselines?

Figure 2:
- Over how many seeds was this run? It would be good to see the confidence intervals for these results.

Figures 4 and 5.
The comparison here is not clear: which images are from the baseline and which are from ComfyGen?

Figure 6:
A larger discussion on the outputs would be beneficial: why do these show that ComfyGen is better? ComfyGen-FT generated two green apples which were not in the prompt, for example.

Additional Experiment Suggestions:
- It would be useful to evaluate how accurate the generation of the ComfyUI JSON is in isolation, since this is the contribution. For example, how much further work is required from the user to ensure that the generated output is parsable by/compatible with the ComfyUI?
- It would be interesting to evaluate the typical workflow(s) produced by the LLM? What types of routes does it construct in particular? Which components/models come up more frequently? How does this relate to the training data; is this something that an expert would have likely created themselves, or has the LLM exploited an unusual pattern.

---

### Official Review · Reviewer_7xXh · 2025-02-28

**Rating:** 6
**Confidence:** 4
**Fit:** 4

**Summary:**

The paper introduces an innovative framework called COMFYGEN, designed to automatically generate text-to-image workflows based on user prompts. This workflow can automatically select and combine the necessary components to significantly improve image generation quality, without requiring users to have expert knowledge of the complex components. The core contribution of the paper lies in proposing two methods that utilize LLMs to generate adaptive workflows: ComfyGen-FT method, which is tuning-based, and the ComfyGen-IC method, which relies on in-context learning. By comparing COMFYGEN to single model approaches, fixed workflows, and other uses of LLMs in generation layout prediction, the research demonstrates that COMFYGEN excels in selecting components suited to the generation task.

**Reason For Giving A Higher Score:**

Please refer to the Strengths.

**Reason For Giving A Lower Score:**

Please refer to the weaknesses.

**Strengths And Weaknesses:**

Strengths
•	The COMFYGEN framework offers a highly practical solution for enhancing text-to-image generation by automating the selection and integration of specialized components. This eliminates the need for users to have deep expertise in understanding the complex interdependencies among different models, making it accessible and useful for a broader audience.
•	The paper is well-written, with clear and concise explanations of both the challenges in text-to-image generation and the proposed solutions.
•	The effectiveness of COMFYGEN has been validated through user studies, adding credibility to its practical applications. This empirical evidence demonstrates that COMFYGEN not only performs well on automated metrics but also meets user expectations and preferences.

Weaknesses
1.	The effectiveness of COMFYGEN is inherently dependent on the quality and diversity of the available specialized components. If the components are not well-optimized or lack diversity, the overall quality of generated images might be limited, potentially affecting the framework's generalization capability across different styles and requirements.
1.	The experiments primarily are conducted on text-to-image generation tasks, leaving questions about the framework's adaptability to other types of generative tasks, such as image-to-image translation. Expanding the scope of evaluation could provide a more comprehensive understanding of the framework's versatility and limitations in different generative contexts.
2.	The paper does not thoroughly analyze scenarios where ComfyGen-IC and ComfyGen-FT might make incorrect decisions in selecting components. Also lacks of discussion on how the use of inappropriate components might negatively impact the quality of the generated images.

**Suggestions:**

1.	Which specific components are used in the COMFYGEN framework? Could you provide details on the specific models and parameter configurations for these components?
2.	Will the fine-tuning data for Llama 3.1 be released in the future?
3.	How does the COMFYGEN framework handle the integration of newly developed components, and is there a mechanism in place for continuously updating the system with the latest advancements in text-to-image generation models?

---

### Official Review · Reviewer_PGSn · 2025-03-03

**Rating:** 6
**Confidence:** 2
**Fit:** 4

**Summary:**

ComfyGen automatically generates adaptive workflows for text-to-image generation using Large Language Models. Instead of relying on a single, monolithic model, it dynamically assembles specialized components based on the user’s prompt. The goal is to improve text-to-image generation quality.

**Reason For Giving A Higher Score:**

I believe that the weaknesses I highlighted are partially caused by the page-limitation and may not be presented in the expanded work. The topic the authors have focused on is important and they explain and show the potential benefits of more prompt-adaptive methods. I believe that the authors are far enough into their work to provide interesting insights at MCDC. I also believe that discussions at MCDC will be beneficial to the authors in future expansion on this work.

**Reason For Giving A Lower Score:**

As mentioned in the weaknesses section the method contains many components and would have benefitted from more explanation and breakdown of components. Additionally, the results would have benefited from more discussion opposed to the more generalised statement under figure 2.

**Strengths And Weaknesses:**

Strengths:

* Prompt-adaptation is an important problem. If we treat all prompts the same the model loses flexibility, being mindful of prompt contents and dynamically adapting the generation workflow could be used to improve outputs.

* In the 2-page space the authors had for the submission they are fairly clear on why prompt-adaption is important and provide good examples within the text.

* I appreciate the idea of using a user interface (ComfyUI) to visualize workflows. When applied correctly, this could offer a user-friendly way for users to tailor their generative systems. However, in this submission, ComfyUI introduced significant complexity with limited expansion on its capabilities.

*  The results show that even when outperformed by baselines, ComfyGen-FT remains comparable. The baselines often experience more extreme drops in results across conditions compared to their counter-parts.


Weaknesses:

* I understand that this is a short-submission but your methodology and contributions feel unclear. Especially when creating workflows or methods that consist of multiple components clear explanations on each component, their method,  and how they interact are needed. I understand this is difficult but perhaps aiming for a 6-page submission would have helped. For example "collected 500 diverse prompts and tested 310 workflows", what was the criteria and how did you collect them? Another example "The LLM analyzes new prompts and matches them to workflows that performed well on similar content.", the idea is good but some expansion would be nice.

*  "ComfyGen-FT outperforms all baseline approaches" feels like an overstatement. In figure 2 we see different baseline approaches outperforming ComfyGen-FT in position, attribute binding, counting, and single object. The only feature that ComfyGen-FT consistently achieves higher performance is for the two object comparison. However, your results are good, they would just benefit from a more concise comparitive statement.

**Suggestions:**

Again, I understand that the 2-page submission length means a fair amount of content is omitted but make sure in the full study you are careful about the statements you make regarding results. Generally, having more prompt adaptive methodology leads to some additional training and infer costs, discussion and analysis of this would be good. The remaining suggestions I have are all mentioned in strengths and weaknesses.

---

### Decision · Program_Chairs · 2025-03-06

**Decision:**

Accept

**Comment:**

This paper expolores the idea of automatically generating text-to-image workflows based on the user prompts. This problem can also be seen as how to route the input throgh different modules and hence it is a good fit for the workshop. Most of the reviewers have a positive opinion on the paper. To further strengthen the paper the paper we recommend to provide more details for the experimental setup and how the data was produced following review hxGf's suggestion. Overall, we recommend accepting this paper.